# Zeta Potential of Nanosilica in 50% Aqueous Ethylene Glycol and in 50% Aqueous Propylene Glycol

**DOI:** 10.3390/molecules28031335

**Published:** 2023-01-30

**Authors:** Marek Kosmulski, Marta Kalbarczyk

**Affiliations:** Department of Electrical Engineering, Lublin University of Technology, Nadbystrzycka 38, 20618 Lublin, Poland

**Keywords:** electrophoresis, stability of dispersion, nonaqueous solvents, ionic surfactants, heat transfer fluids, nanosilica, zeta potential

## Abstract

A sufficient amount of ionic surfactants may induce a zeta potential of silica particles dispersed in water–glycol mixtures of about 100 mV in absolute value. Nanoparticles of silica were dispersed in 50-50 ethylene glycol (EG)–water and 50-50 propylene glycol (PG)–water mixtures, and the zeta potential was studied as a function of acid, base, and surfactant concentrations. The addition of HCl had a limited effect on the zeta potential. The addition of NaOH in excess of 10^−5^ M induced a zeta potential of about −80 mV in 50% EG, but in 50% PG the effect of NaOH was less significant. The addition of CTMABr in excess of 10^−3^ M induced a zeta potential of about +100 mV in 50% EG and in 50% PG. The addition of SDS in excess of 10^−3^ M induced a zeta potential of about −80 mV in 50% EG and in 50% PG. Long-chained analogs of SDS were even more efficient than SDS, but their application is limited by their low solubility in aqueous glycols.

## 1. Introduction

Heat transfer fluids have been extensively studied. They are used to transfer and store heat energy in households, in vehicles, and in power engineering. Low viscosity, high heat capacity per kg, high heat capacity per dm^3^, high heat conductance, broad liquid range, low chemical reactivity, low toxicity, and low cost are the most important factors affecting the design of a heat transfer fluid. In real systems, the above requirements can only be partially met. For example, water has disadvantages of low heat conductance and of narrow liquid range. The liquid range of water can be broadened twice by addition of glycol (at the expense of other advantages of pure water). Mixtures of water with glycerol (and with other polyols) also have broader liquid ranges than pure water, but ethylene glycol is by far the most popular organic co-solvent used in formulations of heat transfer fluids. Both water and aqueous glycol have a low heat conductance, and the heat conductance can be improved by addition of solid nanoparticles, i.e., by using a nanofluid instead of pure solvent. With most nanofluids coagulation is a problem, but the stability against coagulation can be improved by addition of ionic surfactants to the dispersion. Direct studies of long-term stability are tedious, and the zeta potential measurement has been proposed to roughly estimate the stability. High zeta potentials (>100 mV in absolute value) usually imply high stability. In contrast with long-term stability, the measurements of zeta potentials are relatively quick, but they are not trivial, and the interpretation of the results may be difficult, especially in nonaqueous and mixed solvents. Commercial zetameters are designed for aqueous systems, and measurements in nonaqueous media are beyond their standard operation mode. The heat transfer nanofluids must have a high solid load, otherwise the increase in the heat conductance with respect to pure solvents (no particles added) is insignificant. Unfortunately, electrophoresis, which is the most common method used to measure the zeta potential, requires a transparent dispersion; that is, low solid load. The zeta potentials obtained at various solid loads at the same equilibrium concentrations of solutes are consistent. However, the equilibrium concentrations of surfactants are seldom available in the studies of heat transfer fluids, and only initial (total) concentrations of the surfactants are known, but the equilibrium concentrations are not known. With constant initial (total) concentrations of surfactants, the zeta potentials depend on the solid-to-liquid ratio. This problem can be circumvented by dilution of concentrated dispersions with its supernatant. Such an approach has been used in measurements related to mineral processing, but to the best knowledge of the present authors, it has not been used in studies of heat transfer nanofluids.

Several studies of heat transfer fluids offer a simplified approach, in which the stability of dispersions against coagulation is unambiguously defined by the absolute value of the zeta potential of solid particles in that dispersion [1]. Indeed, the zeta potential is correlated with the stability, and in aqueous dispersions, an absolute value of a zeta potential in excess of 60 mV usually assures good stability, and the dispersions are unstable when the absolute value of the zeta potential is below 20 mV. However, this is only a rule of thumb, which is acceptable in popular science, e.g., in Wikipedia, but not in engineering or in serious scientific research. Actually, there is no simple relationship between the zeta potential and the stability: the dispersion stability depends not only on the zeta potential but also on the effective Hamaker constant of the particles (in certain solvents) and on the ionic strength [2]. The typical values of effective Hamaker constant and of the ionic strength in aqueous systems are not necessarily relevant to nonaqueous and mixed solvents.

Conversion of the measured electrophoretic mobilities into a zeta potential requires information on viscosity and dielectric constant of the solvent [3,4,5,6,7]. Their values are well known for water, but they are not always available for nonaqueous solvents, and especially for mixed solvents. The relationship between electrophoretic mobility and zeta potential is straightforward (Smoluchowski or Huckel equation) when the electric double layer is very thin or very thick, but in most real systems, the electric double layer is neither very thin nor very thick, and conversion of the measured electrophoretic mobilities into a zeta potential is complicated even in aqueous systems. An exact solution is only known for monodispersed spheres [8], and for odd-shaped particles we can only estimate the zeta potential. Moreover, even for monodispersed spheres we need extra data (e.g., concentrations and molar conductances of particular ions) on top of viscosity and dielectric constant of the solvent, and such data are usually not available for nonaqueous solvents, and especially for mixed solvents. In view of the difficulties in exact determination of zeta potentials, many investigators used a simplified approach, e.g., a Smoluchowski or Henry equation, which probably underestimated the zeta potentials, at least when they are high in absolute value.

Silica has been considered as a promising component of heat transfer fluids. Silica is inexpensive, environmentally friendly, and is commercially available as particles of various sizes and shapes, including nanoparticles and monodispersed spheres of different sizes. The surface charge of silica can be easily adjusted, and multiple scientific papers describe the pH-dependent surface charging of silica and the effect of surfactants, polymers, and other surface-active compounds on its zeta potential. Moreover, the silica surface can be functionalized. Mukherjee [9] studied dispersions of silica nanopowder in ethylene glycol (EG) stabilized by Arabic gum as a surfactant, at various temperatures. The presence of silica at a concentration as low as 0.5% (by volume) enhanced the thermal conductance by 17–19% at 30–60 °C while the viscosity was only enhanced by 4–5%. Rajendra Prasad [10] studied dispersions of silica nanopowder (30–50 nm in diameter) in aqueous 30% glycerol adjusted to pH 7.7, at various temperatures. The presence of silica at a concentration as low as 1.5% (by mass) enhanced the thermal conductance by 26–32% at 30–70 °C while the viscosity was enhanced by 40–46%.

Akilu [11] summarized seven older studies of dispersions of nanosilica (7–46 nm) in glycol, volume fraction of 0.1–5%, as well as one study in glycerol at room temperature (20–25 °C). The presence of silica enhanced the thermal conductance by up to 25% while the viscosity was enhanced by 31–67%. They also studied dispersions of silica nanopowder (21 nm in diameter) in EG and in glycerol adjusted to pH 10 with NaOH, at various temperatures. The presence of silica at a concentration of 2% (by volume) enhanced the thermal conductance by 6–11% in EG and by 3–5% in glycerol at 30–60 °C, while the viscosity was enhanced by 63–68% in EG and by 19–41% in glycerol.

The above studies by Mukherjee, Rajendra Prasad, and Akilu report on zeta potential measurements in their dispersions. Akilu did not report any specific value; they only state that “absolute zeta potential (…) reached the acceptable stability limit of ±30 mV”. Rajendra Prasad reports a graph zeta vs. pH for 1 d and 30 d-aged dispersions at 30 °C. The effect of aging was rather insignificant. The zeta potential was slightly positive in strongly acidic medium and in strongly basic medium, and a minimum value of −35 mV was reached at pH 7.7. Such behavior is rather unexpected: in aqueous media, the zeta potential of silica is also slightly positive at strongly acidic pH, but at neutral and basic pH the zeta potential of silica is negative [12]. Mukherjee reported the absolute values of zeta potential of silica of about 45 mV, but not the sign. Aging and solid load had rather insignificant effect on the absolute values of zeta potential. In another study, Mukherjee [13] studied dispersions of silica in a commercial coolant (>50% EG) as potential heat transfer fluids. Several studies were also performed [14,15] with heat transfer fluids containing silica and another powder (alumina, titania).

In a series of recent papers [16,17,18], we have demonstrated the possibility of obtaining zeta potentials in excess of 100 mV in absolute value in dispersions of titania and alumina in EG, in propylene glycol, and in their mixtures with water by addition of acid, base, or ionic surfactants. Such high zeta potentials are likely to stabilize dispersions against coagulation, and stability is much desired in heat transfer fluids. Moreover, dispersions with high absolute values of zeta potentials are less viscous that dispersions near the isoelectric point IEP (ζ = 0), and this is another advantage with respect to heat transfer fluids. In this study, we investigate if the zeta potential of silica in 50% aqueous glycols can be adjusted in a similar way. The difference between titania and alumina on the one hand and silica on the other is in location of the IEP on the pH scale in aqueous dispersions. In titania and alumina, the IEP is near the center of the pH scale, so both negative and positive surface charges can be induced by addition of small amounts of acid or base. In contrast, silica is negatively charged over a wide pH range and a positive charge can be only induced by strongly acidic solution.

## 2. Experimental Section

The materials and methods were similar as in our previous studies [17,18]. Fumed silica (99.8%, specific surface area 380 m^2^/g) was from Aldrich (St. Louis, MO, USA) and it was used as obtained. It consists of primary spherical particles about 14 nm in diameter interconnected into pearl-necklace-shaped structures. Reagent-grade ethylene glycol and propane-1,2-diol were from POCh, Gliwice, Poland. The ionic surfactants were from Sigma-Aldrich (St. Louis, MO, USA). The other reagents were from POCh, Lublin, Poland.

Aliquots of 1 kg of 50-50 glycol–water mixture by mass were prepared to assure constant composition in the series of experiments. The stock solutions of SDS and CTMABr in mixed solvents were 5 g/L, and the stock solutions of sodium tetradecyl- and hexadecyl sulfate were 0.5 g/L.

The stock solutions of NaOH and HCl (0.1 M) in 50% glycols were prepared from anhydrous glycols, 1 M aqueous solutions, and water. These stock solutions and pure solvent (50% glycol) were used to prepare dispersions of silica.

The dispersions consisted of 2.5 mg of silica and 25 mL of solution, and they were prepared and sonified just before the measurement. The electrophoretic mobility was determined by means of Malvern Zetasizer at 25 °C. The measurement was repeated 3 times for each dispersion.

The electrophoretic mobility was converted into zeta potential by means of a Henry equation. We arbitrarily assumed a particle radius of 50 nm to calculate the *f* parameter. This value is close to the apparent particle size measured by dynamic light scattering (DLS). The values of particle size produced by the Malvern software were scattered, and their physical sense is not obvious due to the non-spherical shape of the particles. We also emphasize that an exact value of the particle radius is not crucial in calculation of the *f* parameter of the Henry equation. The Henry equation is basically designed for low zeta potentials and for spherical particles, so the values presented here should be considered as an approximation. We still believe that our approximation is more suitable than the Smoluchowski equation for the systems of interest. The following values for 50% EG at 25 °C—viscosity of 3.35 cP, and dielectric constant of 51—and for 50% PG—viscosity of 5.33 cP, and dielectric constant of 57.12—were used to convert the mobility into the zeta potential.

## 3. Results and Discussion

Although the Malvern Zetasizer is not the most recommended instrument to measure the electric conductance of solutions, we used the conductance measurements to assess the degree of dissociation of electrolytes in mixed solvents. The conductance of NaOH and HCl solutions in 50% EG and in 50% PG was proportional to electrolyte concentration over the studied range (correlation coefficient > 0.99), and we believe that these electrolytes are strong electrolytes in 50% aqueous glycols.

Conductance of dispersions of silica (present study) and alumina (our previous study) in 50% EG and in 50% PG containing SDS is plotted in Figure 1. In view of low solid contents (1:10,000), the presence of dispersed particles and their nature had rather insignificant effect on the conductance. Nevertheless, the straight line does not cross the point 0,0 (the same applies to dispersions containing other surfactants). This suggests that the solid particles have a low but measurable contribution to the conductance of the dispersions. The line connecting the points in Figure 1 is straight over the studied concentration range for 50% EG and for 50% PG, and this suggests that the CMC is beyond the experimental range, that is, >0.017 M. This figure is substantially higher than the CMC of SDS in water (0.008 M), and this is in line with the results reported by the others, namely, the CMC of SDS increased on addition of organic co-solvent to water [19].

The slope of the straight line connecting the points for 50% EG is higher than the corresponding slope for 50% PG, and the ratio of the slopes is proportional to reciprocal viscosity of the solvent.

Conductance of dispersions of silica (present study) and alumina (our previous study) in 50% EG and in 50% PG containing CTMABr is plotted in Figure 2. Unlike in Figure 1, the plots’ conductance vs. concentration is not linear. We speculate that the curvature is due to micellization, but it can also be due to ion-pairing (weak electrolyte). The discrepancies between the results obtained with silica and alumina in the range of high surfactant concentrations suggests that, along with micellization, we deal with structures involving both surfactant molecules and solid particles.

The rectilinear portion of the conductance vs. concentration curves extends up to 0.003 M in 50% EG and up to 0.006 M in 50% PG, and we speculate that these figures may indicate the CMC of CTMABr in both solvents. These figures are substantially higher than the CMC of CTMABr in water (0.0009 M), and increase in the CMC of CTMABr induced by addition of organic co-solvent to water is well-known [20]. The slope of the tangent to the low-concentration segment of the line connecting the points for 50% EG in Figure 2 is higher than the corresponding slope for 50% PG, and the ratio of the slopes is proportional to reciprocal viscosity.

We conducted similar analysis of the conductance vs. concentration curves with long-chained analogs of SDS, but the results are not unequivocal due to a narrow concentration range (low solubility), low molar conductance, and thus substantial contribution of solid particles to the conductance as compared with the contributions of the surfactants. Apparently, our experiments were carried out in the concentration range of surfactants exceeding the CMC for CTMABr and below the CMC for other surfactants. The measurement of electric conductance alone is not the most recommended method for determination of the CMC, but linear dependence of conductance on concentration suggests that we are below the CMC in the case of SDS, and with its long-chained analogs, there is no micellization at all.

The possibility of receiving highly charged silica particles by means of addition of acid or base in 50% EG is illustrated in Figure 3. The data points near the left axis refer to a dispersion without acid or base added. The error bars indicate that the standard deviation in the zeta potential is often in excess of 10 mV.

The addition of acid is capable of sign reversal of zeta potential to positive, but the absolute values of the positive zeta potential are too low to assure electrostatic stabilization of dispersion. In this respect, the behavior of dispersion in 50% EG is similar to the properties of aqueous dispersions of silica: positive zeta potentials can be induced by acidification, but the absolute values of the zeta potential are low. The results obtained by different authors in aqueous dispersions are contradictory, and several studies report negative zeta potentials of silica even at pH about 0.

The addition of a base enhanced the original negative zeta potential from −40 (no acid or base added) to about −80 mV. A maximum negative zeta potential of −80 mV was reached in 10^−5^ M NaOH, and further addition of a base had rather insignificant effect. In this respect, the electrokinetic behavior of silica is similar to that in aqueous dispersions, namely high negative zeta potentials of silica are observed at pH as low as 7, and further increase in pH does not enhance the negative zeta potential. Negative zeta potential of −80 mV in basic dispersions may be sufficient for electrostatic stabilization of a dispersion, but −80 mV is rather discouraging as compared with negative zeta potentials in excess of −100 mV reported in our previous papers for nanotitania and nanoalumina.

The attempts at obtaining high absolute values of zeta potentials by addition of acid or base were even less successful for 50% PG. The zeta potential of silica in 50% PG (no acid or base added) is positive (Figure 4) and it differs substantially from the zeta potential of silica in 50% EG (no acid or base added), which is negative (Figure 3). This difference is most likely due to small amounts of surface-active impurities in the reagent-grade chemicals. The nature and concentration of these impurities can be very different in spite of the chemical similarity of both glycols. Moreover, the nature and concentration of impurities varies from one lot of the same solvent to another. Hypotheses have been coined, which relate the zeta potential of particles in a solvent (no solutes added) to the properties of the solvent (donor number, acceptor number), but the present authors are skeptical about such hypotheses. The addition of acid had rather insignificant effect on the zeta potential of silica in 50% PG. The addition of a base is capable of sign reversal of zeta potential to negative, but the absolute values of the negative zeta potential are too low to assure electrostatic stabilization of the dispersion.

The possibility of receiving highly charged silica particles by means of addition of SDS or CTMABr in 50% EG is illustrated in Figure 5. SDS had rather insignificant effect on the zeta potential, which was shifted from about −40 mV (pure solvent) to about −80 mV (10^−4^ M SDS), and further addition of SDS up to 10^−2^ did not induce further increase in the negative zeta potential. About 10^−5^ M CTMABr was sufficient to reverse the sign of the zeta potential to positive, and increase in the CTMABr concentration induced a positive zeta potential, which reached +100 mV at 10^−3^ M CTMABr, but further increase in the CTMABr concentration up to 10^−2^ did not induce an increase in the positive zeta potential. Apparently, CTMABr is more efficient than SDS as an agent at inducing high zeta potential of silica (in absolute value), and thus improving the stability of dispersions and depressing their viscosity. We observed similar trends with titania and alumina; that is, CTMABr was superior to SDS in terms of high zeta potential in 50% EG (in absolute value).

Dispersions of silica in 50% PG behaved differently from dispersions in 50% EG, as illustrated in Figure 6. Addition of SDS at concentrations up to 5 × 10^−4^ M had rather insignificant effect on the zeta potential of silica, but an increase in SDS concentration from 5 × 10^−4^ to 10^−2^ M resulted in systematic increase in the negative zeta potential up to −140 mV. The latter value is comparable in absolute value with the highest positive zeta potentials in 50% PG induced by CTMABr. Apparently, both SDS and CTMABr, at concentrations of about 10^−2^ M, can induce high zeta potential of silica (in absolute value), and both surfactants can be used to produce stable dispersions of low viscosity in 50% PG.

The effect of the chain length in sodium alkyl sulfates on the zeta potential of silica in 50% EG is presented in Figure 7. The range of experimental data for sodium tetradecyl- and hexadecyl sulfate is limited by their solubility in 50% EG. Apparently, an increase of the chain length from C12 (SDS) to C14 had rather insignificant effect on the zeta potential. On the other hand, the zeta potentials obtained in sodium hexadecyl sulfate were substantially more negative than in SDS, and in >10^−4^ M sodium hexadecyl sulfate the absolute value of zeta potential was even higher than the maximum absolute value observed in CTMABr (Figure 5). These results suggests that sodium hexadecyl sulfate is more suitable than SDS to obtain stable and low-viscous dispersions of silica in 50% EG.

Figure 8 shows that the effect of chain length in sodium alkyl sulfates on the zeta potential of silica in 50% PG is rather insignificant. Although sign reversal of zeta potential from positive (no surfactant added) to negative was observed at sodium hexadecyl sulfate concentrations below 10^−5^ M, further addition of sodium hexadecyl sulfate did not induce high negative zeta potentials sufficient to stabilize a dispersion.

Finally, we emphasize that in terms of correlation between dispersion stability and zeta potential, silica differs substantially from metal oxides in aqueous dispersions, and it can also be the case in mixed (aqueous–organic) solvents. For example, Matijevic [21] studied the stability of silica aqueous dispersions in various electrolytes over a wide range of ionic strengths and pH. No correlation was found between zeta potential and stability. In this respect, the preliminary studies of zeta potential aimed at assessment of dispersion stability [1] need to be confirmed by direct studies of stability for silica even to a higher degree than for other oxides.

## 4. Conclusions and Further Research

Unlike with titania and alumina, acids and bases are not suitable to induce zeta potentials of silica in aqueous glycols sufficient to stabilize a dispersion. In contrast, highly charged particles can be produced by the addition of CTMABr. We only worked with 50% aqueous glycols, but we speculate that the present results are relevant to a broad range of glycol concentrations (e.g., 40 or 60% EG or PG). The problem with fumed silica is in conversion of electrophoretic mobility into zeta potential for pearl-necklace-shaped aggregates. Similar experiments with monodispersed spherical particles (Stober silica) would result in more credible values of the zeta potential. Functionalization of silica may be a more efficient solution than addition of surfactants. The results of this study are useful in the formulation of heat transfer fluids for practical applications, based on silica alone or on silica along with other solid particles. Positively charged silica dispersions produced by addition of CTMABr are especially promising.

## Figures and Tables

**Figure 1 molecules-28-01335-f001:**
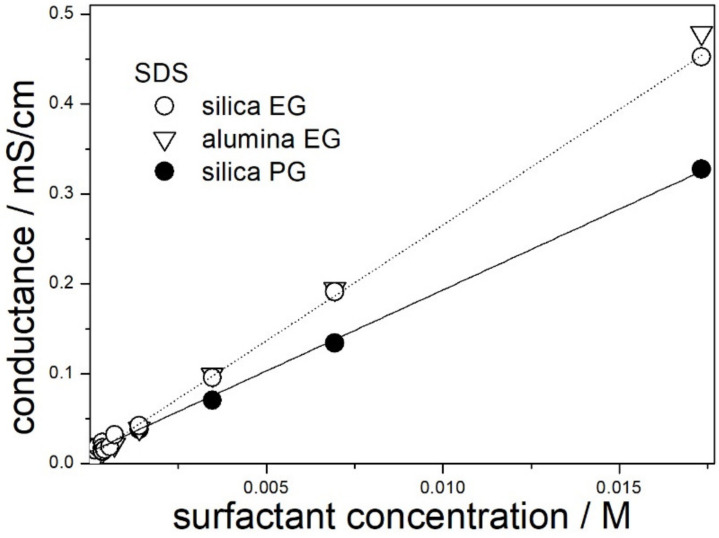
Electric conductance of dispersions in 50% EG and in 50% PG containing SDS.

**Figure 2 molecules-28-01335-f002:**
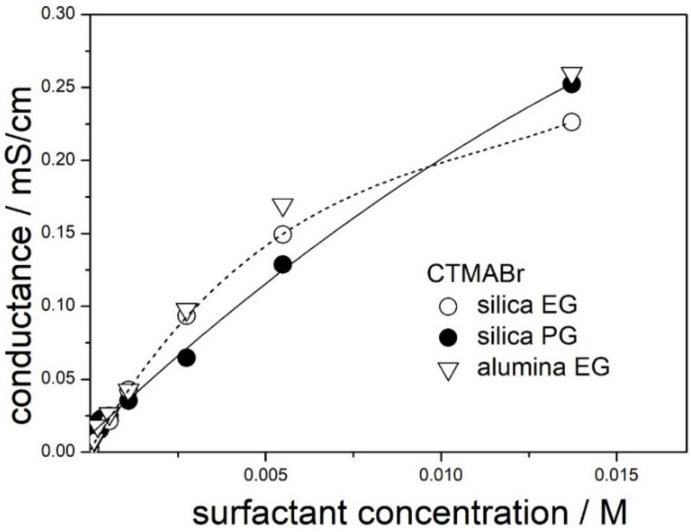
Electric conductance of dispersions in 50% EG and in 50% PG containing CTMABr.

**Figure 3 molecules-28-01335-f003:**
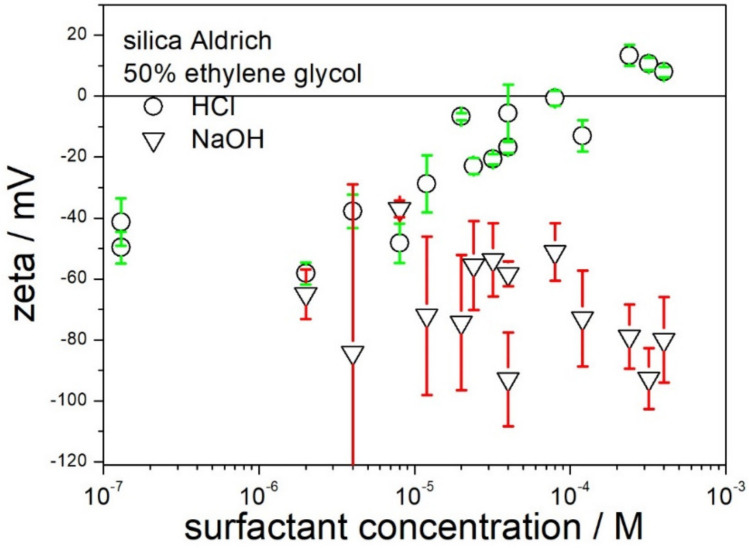
The effect of acid and base on the zeta potential of silica in 50% EG.

**Figure 4 molecules-28-01335-f004:**
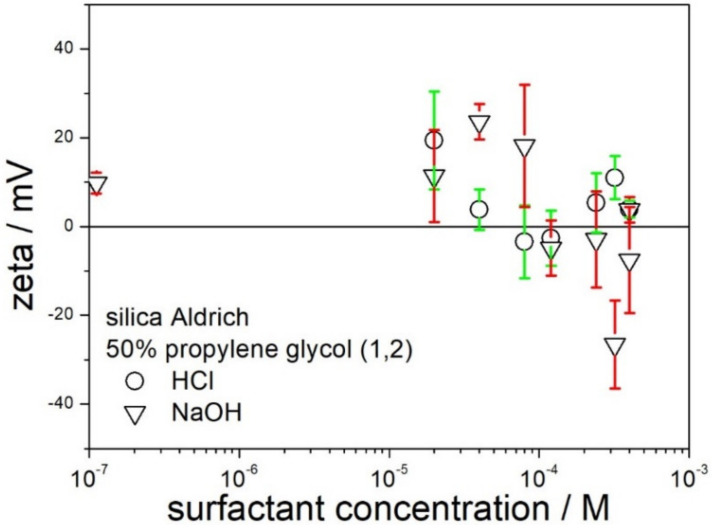
The effect of acid and base on the zeta potential of silica in 50% PG.

**Figure 5 molecules-28-01335-f005:**
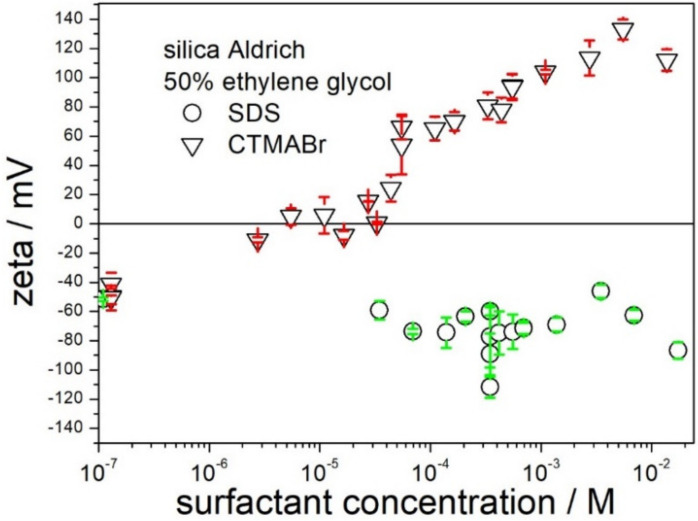
The effect of SDS and CTMABr on the zeta potential of silica in 50% EG.

**Figure 6 molecules-28-01335-f006:**
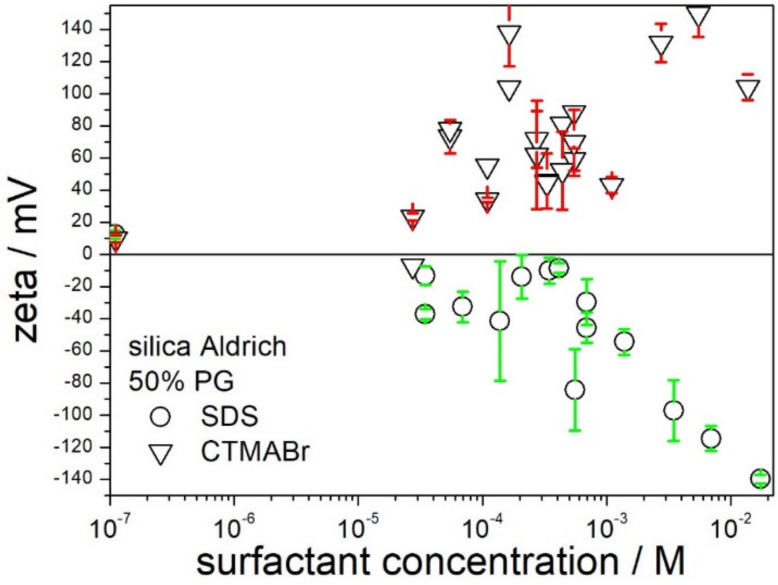
The effect of SDS and CTMABr on the zeta potential of silica in 50% PG.

**Figure 7 molecules-28-01335-f007:**
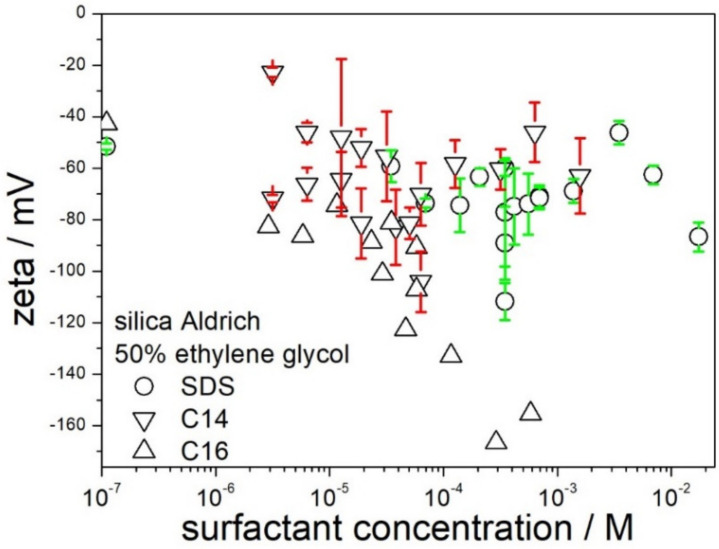
The effect of SDS, sodium tetradecyl- and hexadecyl sulfate on the zeta potential of silica in 50% EG.

**Figure 8 molecules-28-01335-f008:**
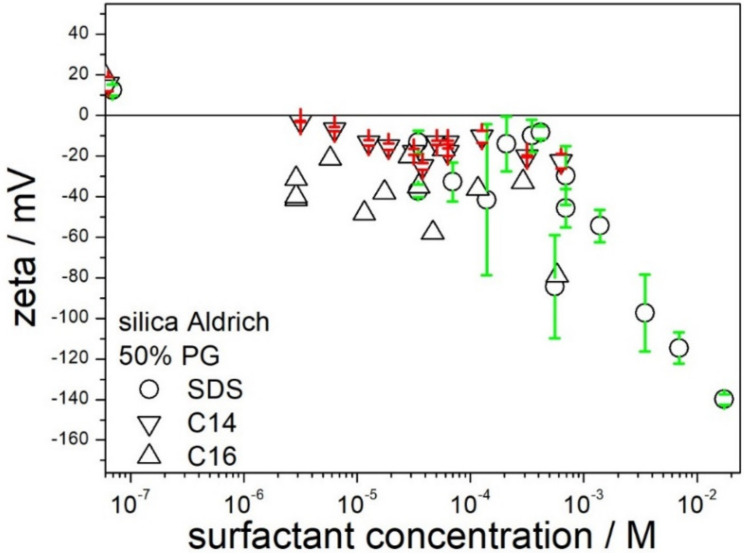
The effect of SDS, sodium tetradecyl- and hexadecyl sulfate on the zeta potential of silica in 50% PG.

## Data Availability

Data are available from the corresponding author on request.

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
