# Peer review of "Zeta Potential of Nanosilica in 50% Aqueous Ethylene Glycol and in 50% Aqueous Propylene Glycol"

_molecules, 2023, doi:10.3390/molecules28031335_

Round 1

Reviewer 1 Report

1.The title is not catchy and does not reflect essential contents.
2.Where is the practical application of this manuscript? It must be added.
3. Check the grammar throughout the article and correct it. Proofread the article as many language errors were identified.
4. The conclusion must be more than just a summary of the manuscript.
5. Overall, the keywords must be rewritten
6. The authors need to follow the author's guidelines when they writing the manuscript.
7.The introduction needs to be further revised to highlight the purpose of the study, You need to introduce what others have studied and what needs further research. Besides, the following all of references are recommended to be cited:
https://www.sciencedirect.com/science/article/abs/pii/S0360319922008126
https://ceramics.onlinelibrary.wiley.com/doi/abs/10.1111/jace.17696
https://www.sciencedirect.com/science/article/pii/S0167732221011296

Author Response

1.The title is not catchy and does not reflect essential contents.

The title was changed to more specific.

Zeta potential of nanosilica in 50% aqueous ethylene glycol and in 50% aqueous propylene glycol

2.Where is the practical application of this manuscript? It must be added.

The following was added in the intro.

They are used to transfer and store heat energy in household, in vehicles, and in power engineering.

The following was added in the conclusion.

The results of this study are useful in formulation of heat transfer fluids for practical applications, based on silica alone or on silica along with other solid particles. Positively charged silica dispersions produced by addition of CTMABr are especially promising.

  1. Check the grammar throughout the article and correct it. Proofread the article as many language errors were identified.

The text was checked and corrected by a native speaker.

The manuscript contains many specialistic expressions, which are not in the dictionary. Therefore the spell checker detects many “errors”.

  1. The conclusion must be more than just a summary of the manuscript.

The following was added in the conclusion.

The results of this study are useful in formulation of heat transfer fluids for practical applications, based on silica alone or on silica along with other solid particles. Positively charged silica dispersions produced by addition of CTMABr are especially promising

  1. Overall, the keywords must be rewritten.

The keywords were rewritten.

  1. The authors need to follow the author's guidelines when they writing the manuscript.

We used a newly launched option which is “free format”.

https://www.mdpi.com/journal/molecules/instructions

Our Ms was converted to the Molecules template by the editorial office.

7.The introduction needs to be further revised to highlight the purpose of the study, You need to introduce what others have studied and what needs further research. Besides, the following all of references are recommended to be cited:

https://www.sciencedirect.com/science/article/abs/pii/S0360319922008126
https://ceramics.onlinelibrary.wiley.com/doi/abs/10.1111/jace.17696
https://www.sciencedirect.com/science/article/pii/S0167732221011296

We would gladly add new references. However:

We have carefully studied the above papers and

They are not about silica

They are not about heat transfer fluids

They are not about glycols or about their mixtures with water

They are not about zeta potential.

Therefore they are not relevant to our study and we will not cite them.

Reviewer 2 Report

The primary issue raised by the study is whether silica's zeta potential in 50% aqueous glycols may be altered. The subject seems fresh and current in the field. Additionally, it fills a particular knowledge gap. This article offers more light than others on the precise mechanisms that allow modulation of the zeta potential of mixes of nanosilica dispersed in aqueous glycols. The authors should consider some explanations regarding the methodology, answering the question of why they did not use dynamic light scattering methods for estimation of zeta potential values. Currently, the gold standard is considered to be dynamic light scattering methods. 

The conclusions do address the primary question raised and are consistent with the arguments and evidence presented. The references are appropriate. The text is sloppily written. Not all abbreviations are deciphered, which makes it impossible or significantly difficult to understand what is written. For the convenience of the eye, the trend lines in the figures should be presented.

The authors have to strive to improve the quality of the text, including the English grammar.

Author Response

The primary issue raised by the study is whether silica's zeta potential in 50% aqueous glycols may be altered. The subject seems fresh and current in the field. Additionally, it fills a particular knowledge gap. This article offers more light than others on the precise mechanisms that allow modulation of the zeta potential of mixes of nanosilica dispersed in aqueous glycols.

1.The authors should consider some explanations regarding the methodology, answering the question of why they did not use dynamic light scattering methods for estimation of zeta potential values. Currently, the gold standard is considered to be dynamic light scattering methods. 

We believe that the referee meant: “why they did not use dynamic light scattering methods to determine the particle size (which can be further used to estimate the zeta potential from electrophoretic mobility)”.

This is a very good point, but we do not like to discuss it in too much detail in our paper.

We agree that using the particle size from DLS is superior to using the size of primary particles, but even in specialistic journals devoted to colloid chemistry, the former approach is rare. Most authors use either Smoluchowski or more sophisticated methods, but with the size of primary particles.

Most scientists engaged in heat transfer fluids are not familiar with the problem of conversion of electrophoretic mobility into zeta potential and they use Smoluchowski equation. With spherical particles we can use more sophisticated methods, e.g., O Brien-White. The problem is that even the primary particles are not spherical in our system. The estimation of particle size by DLS is based on an assumption that the particles are spherical. The analysis of autocorrelation function will always produce some “effective particle size” but the significance of such effective size is not clear when the primary particles are not spherical. The situation is even more complex in the presence of aggregates. Not only the aggregates are non-spherical, but they are also porous and dynamic. Not surprisingly the particle size measured in our systems was irreproducible. However the effect of the particle size on the calculated zeta potential is not very significant. Considering that the scatter of results was on the order of 10 mV, the potential error induced by taking “wrong” value of particle size is minor (unless the error is by an order of magnitude).

The problem raised by the referee is important, but we do not have sufficient data and we do not know how to interpret existing data. Therefore we only added one sentence:

This value is close to the apparent particle size measured by dynamic light scattering DLS.

  1. The text is sloppily written. Not all abbreviations are deciphered, which makes it impossible or significantly difficult to understand what is written.

The text was carefully revised in order to the abbreviation explanation. The list of used abbreviations was extended.

  1. For the convenience of the eye, the trend lines in the figures should be presented.

The trend lines were shown in Fig. 1 and 2 where the trend is clear. We do not add  trend lines in the other figures. With many symbols and error bars these figures are overcrowded already. Moreover several sets of results are very scattered and we are not sure what the trend lines should look like.

  1. The authors have to strive to improve the quality of the text, including the English grammar.

The text was checked and corrected by a native speaker.